# OpenReview forum: "Large Language Models Develop Novel Social Biases Through Adaptive Exploration"
_ICLR.cc/2026/Conference — Submitted to ICLR 2026_

### Official Review · Reviewer_kwFn · 2025-10-31

**Soundness:** 2
**Presentation:** 3
**Contribution:** 2
**Rating:** 2
**Confidence:** 4

**Summary:**

This paper analyzes the biases present in LLMs, particularly those generated in multi-turn simulations. Different metrics are proposed to analyze a wide range of experimental results. The authors also investigate the forms that biases and explore mitigation methods.

**Strengths:**

1. The multi-turn scenarios that the authors attempt to explore have not been widely studied, which could be regarded as a novel research topic.

2. The authors use multiple LLMs and attempt to evaluate them using different metrics.

**Weaknesses:**

1. The authors compare the simulation results of LLMs with those from human participants, but lack descriptions of the human participants, such as the sample size and distribution of demographic variables.

2. The authors need to provide more explanation for the three newly defined metrics. For example, how do SI and mutual information differ in form? What are the similarities and differences among BGD, GASI, and JSD?

3. If I understand correctly, the values of SI and BGD should be 0 under random conditions, but this doesn't seem to be the case in Figures 2, 4 and 5.

4. There are still some points that are not easy to understand, please refer to the questions section below.

Minor issue: The Figures in Appendix B are difficult to read.

**Questions:**

1. If the same person or multiple individuals' information is reused for prompts, how will the results differ across different rounds?

2. The authors use the default temperature in their simulations. Would the conclusion change if the temperature are set to the maximum or minimum?

3. Does the width of the human data band in Figure 2 represent the standard deviation? If not, what is their standard deviation?

4. In the era before LLMs, biases may also be amplified in multi-turn evaluations?

---

> ### Author Response · Authors · 2025-11-20
> **Response to Reviewer kwFn (1/2)**
>
> We thank the reviewer for their helpful comments and for recognizing the novelty of our work for understanding novel biases emerging from LLM multi-turn interactions. We address each of the reviewer’s points below:
>
> ## **Point 1: Human participants**
>
> We agree that describing the demographics comprising our human baseline will help solidify our findings. Their paper states the following details for human data:
>
> - 1310 participants were sourced from the CloudResearch High-Quality Subject pool (cloudresearch.com). All speak English as their first language and are >18 years old (avg. age 40).
>
> - 51% of the participants were female, 46% male, and 1% non-binary.
>
> - 74% of participants were White, 10% Black, 6% Hispanic, 5% Asian, and 4% multiracial.
>
> - 75% of participants hold some college degree.
>
> - The average political orientation of the participants was 3.94 (1 = extremely conservative, 6 = extremely liberal).
>
> These demographics reflect typical characteristics of human participants in online psychology studies. Crucially, the stratification result in Bai et al. holds when controlling for individual differences in age, gender, race, education, and political orientation (p < 0.001). Of these 1310 participants, 600 were relevant to our baselines: 200 for the classic setting, 200 for the altered setting with p=0.1, and 200 for the diversity steer intervention.  We have included these details in the new Appendix C of the revision.
>
> ## **Point 2: Explanation of metrics**
>
> **SI vs. Mutual Information**
>
> Thanks to the reviewer’s feedback, we have recognized that, with assumptions valid within our controlled setting, SI is equivalent to mutual information—providing additional theoretical grounding for this measure. We provide a new proof of equivalence in Appendix B.1 of the revision.
>
> In our paper, the concept we want to measure is stratification relative to an equal-opportunities baseline. SI, as currently formulated, is easily interpreted as the difference in entropy between the ideal uniform distribution and the actual allocation, and makes the normative baseline of uniformity (the first term) explicit. Thus, we keep the current presentation, but directly mention the equivalence to mutual information in newly revised Section 3.2.
>
> **BGD, GASI, and JSD**
>
> Both BGD and GASI rely on JSD as the fundamental distance measure. JSD quantifies how different two probability distributions are, while BGD and GASI are averages of JSD values taken over specific pairs of distributions. Because of this, all three are bounded, symmetric, and interpretable in the same range [0,1].
>
> BGD measures how different groups are from one another within runs, whereas GASI measures how consistent each group’s behavior or allocation is across runs. Specifically, BGD takes the mean JSD across all paired groups within each run, and then averages across runs. GASI takes the mean JSD across runs for each group, and then averages across groups. These measures offer complementary insights into both the degree of bias, and whether this bias is a result of structural dynamics or static priors.
> | Metric | Varies Over | Holds Constant | Captures |
> |-|-|-|-|
> | **JSD** | No structure (two distributions) | — | Direct distributional difference |
> | **BGD** | Pairs of groups ((g_1, g_2)) | Same run (r) | Between-group separation |
> | **GASI** | Pairs of runs ((r_1, r_2)) | Same group (g) | Across-run stability |
>
> ## **Point 3: Metrics under random sampling case**
>
> In our random baseline, we run 30 trials of 40 job assignments, in which each job is independently randomly assigned to one of the four demographics. The BGD and SI are then computed by averaging across the 30 trials. This represents an allocation that is completely devoid of bias but still contains natural variation, which will exist in any allocation. Thus BGD and SI will be greater than 0.
>
> To contrast, these values would only both be 0 in the (impossible) case in which each demographic is hired 10 times, and assigned to each job quadrant 2.5 times. Then the entropy of the job assignments for each demographic would match the uniform distribution, and the divergence between groups would be 0.
>
> As the number of observations and job assignments per trial increases, natural variations diminish and SI converges toward 0, as desired under randomness (Figure 9 in App. B for SI, Figure 11 for BGD).
>
> ## Minor Issue: Appendix B matrices too small to read
> We have revised this appendix with a writeup that clarifies its purpose: a visualization showing that more advanced models stratify more strongly (Appendix E, paragraph 1). Each plot is the rank-ordered allocation matrix for a single run. For each of six model families, we illustrate how the matrices for 30 runs go from more uniform (e.g. Claude 3 Haiku) to nearing the identity matrix (e.g. 4 Sonnet) as models become more capable, reflecting our finding that stratification increases with model capability even in the absence of demographic differences.

---

> ### Author Response · Authors · 2025-11-20
> **Response to Reviewer kwFn (2/2)**
>
> ## **Question 1: Prompting Clarification**
>
> To clarify, Tufa, Weki, Aima, and Reku refer to fictional demographic groups, not individuals. Each round presents the LLM with four new applicants, one per each demographic group. No personal information is used, and no individual appears more than once. This is explicitly stated in Section 3.1: “applicants from 4 unfamiliar demographic groups: Tufa, Aima, Reku, and Weki”, as well as in the prompts for LLMs (App. A1).
>
> The LLM’s chain-of-thought traces also provide evidence for this interpretation, consistently reasoning that they are evaluating different individuals from each group:
> > “Previously, the Aima community was identified as having attributes well-aligned with political roles, such as a focus on leadership and governance. Choosing a candidate from the Aima community seems to align with these strengths, making them a suitable fit for a political role.”
>
> ## **Question 2: Analyzing Temperature**
>
> In our original Section 5.1, we provided analyses for higher temperature (T=1.5) to encourage exploration, where we found that with direct prompts, GPT-4o does not produce more diverse allocations with higher temperature.
>
> In our revision, we expand this analysis to the 4 other frontier models (Claude, Gemini, Llama, Qwen), and also to both higher and lower temperatures (T=0, 1.5). At higher temperatures (even T=1.2), CoT outputs were no longer sensible in our multi-turn paradigm, so we only report direct prompting results. As shown in the table below, temperature has minimal effects on stratification.
>
> | Model| Prompting | T0  | T1  | T1.5 |
> |-|-|-|-|-|
> | **Llama 4 Maverick** | CoT |1.36 |1.26 | —   |
> |  | direct |1.19 |1.39 |1.37 |
> | **Qwen 2.5 Instruct**| CoT |1.13 |0.89 | —   |
> | | direct |1.13 |1.08 |0.91 |
> | **Claude Sonnet 4**  | CoT |1.53 |1.66 | —   |
> | | direct |1.67 |1.79 |1.79 |
> | **Gemini 2.5 Flash** | CoT |1.55 |1.53 | —   |
> | | direct |1.65 |1.51 |1.61 |
> | **GPT-4o** | CoT  |1.05 |1.04 | —   |
> | | direct |1.46 |1.30 |1.21 |
>
> ## **Question 3: Human Intervals (Fig 2)**
>
> The bands in Figure 2 represent 95% confidence intervals (SI CI: 0.77–0.89; BGD CI: 0.53–0.57) for the human data (n = 203), following the figure title. The standard deviations are 0.30 and 0.12 respectively, approximately 3 times the CI widths, which is expected for a sample size of ~200.
>
> ## **Question 4: Before LLMs**
>
> We agree. In paragraph 2 of the introduction, we mentioned a few examples across different fields of science where humans amplify their biases from exploration, such as predictive policing and neighborhood segregation. A central contribution of our paper is that LLMs are also susceptible to such bias formation when incorporated into multi-turn decision making paradigms. Importantly, LLMs demonstrate this tendency even more strongly than humans, which can lead to potentially harmful societal impacts if carelessly adopted.
>
> &nbsp;
>
> Once again, we thank the reviewer for their comments and suggestions! Please feel free to respond with any follow-up questions if any concerns still remain.

---

> > ### Author Response · Authors · 2025-11-26
> > **Thank you for your review!**
> >
> > Dear Reviewer kwFn,
> >
> > We greatly appreciate your attention to the methodological aspects of our work and for suggesting further comprehensive experiments, which has greatly helped us reinforce and validate our results and conclusions.
> >
> > As the rebuttal process is ending in just under one week, we would like to know if we have adequately addressed your concerns and questions raised in your original review. If not, we would be more than willing to engage in further discussion.
> >
> > Sincerely,
> >
> > The Authors

---

### Official Review · Reviewer_vqkN · 2025-11-01

**Soundness:** 4
**Presentation:** 4
**Contribution:** 4
**Rating:** 8
**Confidence:** 4

**Summary:**

The authors propose to mention the potential of LLM developing novel social biases about artificial demographic groups through interaction. The authors directly compare the LLM results with the human results and find that LLMs can develop these biases even when no inherent differences exist. They then propose a serious of interventions.

**Strengths:**

The paper is very well-motivated: lots of work have looked at how LLMs could be biased due to fundamental human data distribution but there’s little work in exploring how LLMs might form novel biases through interactions. I like that the authors engage with the literature across many fields with a good amount of depth in making the arguments and describing the background of this study. It is also very nice to have human baseline in a directly comparable setting.

**Weaknesses:**

Weaknesses:

1)	The study models agentic behavior using a multi-turn dialogue where the entire history is passed in-context. This setup, while controlled, does not fully capture the architecture of modern agentic systems. Such systems often employ more sophisticated mechanisms like structured memory, explicit reflection steps (e.g., ReAct), and meta-cognitive abilities to decide whether a given experience is valuable enough to be integrated into its knowledge base. By "forcing" the model to learn from every turn via in-context learning, the experiment may be inadvertently creating a scenario that is highly conducive to the over-generalization it observes. The degree to which these biases emerge in agents with more robust memory and reflection capabilities remains an open question.

2)	Newer and larger models have a greater tendency to stratify -> an explanation is simply that larger model learn better in context?
Also it might be better to present the same result in Figure 3 by plotting the stratification index against standard capability benchmarks (e.g., MMLU, Arena). I suppose with figure 3 the point you are really making is how the stratification tendency changes with model capability, right?

3)	To what degree the result of this study generalize to more real-life cases? I get that in order to avoid measuring existing biases and establish internal validity, you have to create an artificial city with artificial group labels? But then because the LLM also clearly knows this is an artificial setting, it might just act without the normative constraints that it might apply to real demographic groups? In other words, perhaps you are testing bias formation in a “jail-breaked” setting?

4)	Regarding the hair color and tattoo shape result is line 376, do you have any evidence to indicate that these are indeed spurious signals? These features might be correlated with sociodemographic features that might be important?

5)	Just out of curiosity, how do you think your result interact with pre-existing bias, in a more realistic setting? If the task used real demographic groups. would the models lock onto existing stereotypes even faster? Or perhaps the random evidence in context actually reduce the pre-existing model bias?

**Questions:**

see above

---

> ### Author Response · Authors · 2025-11-20
> **Response to Reviewer vqkN (1/2)**
>
> Thank you so much for taking the time to review our paper! We’re very glad you found our work well-motivated in studying novel biases through interactions, and we appreciate your comment about our paper engaging with a breadth of literature across many fields. We would like to take this time to respond to your points.
>
> ## **Point 1: Generalization to agentic systems**
> > **“The degree to which these biases emerge in agents with more robust memory and reflection capabilities remains an open question.”**
>
> Yes, we agree that this is a very important question to answer! Following the reviewer’s suggestion, we have conducted a prototype experiment augmenting LLMs (Gemini 2.5 Flash, GPT-4o) with the ReAct [1] framework commonly used in LLM agents [2]. We find that segregative behaviours persist very strongly even with this agentic addition, as shown in the table below. Although LLM agents may still employ more sophisticated memory and meta-cognitive mechanisms, our findings suggest that the core dynamics of feedback-induced bias formation remain a concern even with agentic scaffolding. See Appendix I for implementation details.
>
> ## **Point 2: Newer models segregate more strongly**
> > **“Newer and larger models have a greater tendency to stratify -> an explanation is simply that larger models learn better in context?”**
>
> It’s true that larger models are better at learning in context, and this certainly leads to greater stratification. However, part of our paper’s contribution is to ask the reader to think critically about what improved learning brings us: Namely, that it does not always lead to better outcomes. We make this explicit in the last paragraph of our introduction, emphasizing the importance of proper objectives (e.g., ones that incorporate societal values) to match stronger LLM capabilities.
>
> > **“it might be better to present the same result in Figure 3 by plotting the stratification index against standard capability benchmarks (e.g., MMLU, Arena)”**
>
> This is a great idea. We have included this comparison in Appendix D using data from the HELM benchmark [3], and reference it in results section 4.2 in blue. As the reviewer predicts, we observe a nice trend that more capable models perform better on BBQ, but are simultaneously more susceptible to developing emergent biases in our experiments.
>
> ## **Point 4: Are the spurious signals correlated with socio-demographic notions?**
>
> We thank the reviewer for their careful attention to our experimental validity. The main source informing our choice of the features hair color and tattoo shape was Martin et al. (2014) [4], indicating these attributes were included as plausible, but not necessarily dominant, visual cues available to decision-makers under uncertainty. Further, age and education are the two most salient features in the reallocation setting [5], so even if hair color and tattoo shape are correlated with important demographics, they would likely still be auxiliary in comparison. We have slightly reduced the claims to focus our results more on comparing the relative validity between different types of information rather than their absolute magnitudes of importance (see Section 5.2).
>
> ## **Point 5: Pre-existing biases**
>
> This is a very interesting question! Instead of providing speculations, we thought it would be better to test directly.
>
> We conducted the hiring experiment using real racial groups (White, Black, Asian, Hispanic) and real jobs that are associated with these groups [6] (see Appendix J for details). In our results below, while we observe similarly high levels of segregation (with SI 1.21–1.52), we remark that these allocations primarily reflect pre-existing priors, evidenced by the much lower GASI values of models’ allocations (0.14–0.30). Thus, it’s likely that as the reviewer guessed, LLMs lock onto existing stereotypes even faster, though it remains to be seen whether this result varies across domains, success rates, and other factors.
>
> | Model| Prompting | SI  | BGD  | GASI |
> |-|-|-|-|-|
> | GPT-4o| Direct | 1.52 | 0.75 | **0.14**
> | | CoT | 1.21 | 0.65 | **0.28**
> | Gemini 2.5 Flash | Direct | 1.41 | 0.72 | **0.22**
> | | CoT | 1.29 | 0.69 | **0.30**
>
> It’s also worth mentioning that our resettlement setting in Section 5.2 did use real demographics, but these were obscure indigenous groups from Central and East Asia: Tofa, Ket, Udi, and Taz. LLMs did still possess knowledge on these groups, and allocations were still very stratified (See Appendix F), but these allocations did not reflect any priors, as shown by relatively high GASI scores (0.43–0.59).

---

> ### Author Response · Authors · 2025-11-20
> **Response to Reviewer vqkN (2/2)**
>
> ## **Point 3: Generalization**
> > **“because the LLM also clearly knows this is an artificial setting, it might just act without the normative constraints that it might apply to real demographic groups?”**
>
> While our original hiring paradigm is artificial to replicate stimuli from the psychology experiment, this allowed us to compare LLM results against humans. In follow-up experiments, we create much more realistic scenarios, testing two new allocation settings (conscription and resettlement) without gamified rewards (see updated Section 5.2, blue paragraph). We also tested real demographics in our response to point 5. In all these conditions LLM still created stratified societies, suggesting that our findings are not limited to only “jail-breaked” settings.
>
> &nbsp;
>
> [1] Yao et al., ReAct: Synergizing reasoning and acting in language models. ICLR, 2023.
>
> [2] smolagents https://github.com/huggingface/smolagents.
>
> [3] Liang et al., Holistic evaluation of language models. TMLR 2023. BBQ URL: https://crfm.stanford.edu/helm/safety/latest/#/leaderboard/bbq
>
> [4] Martin et al., The spontaneous formation of stereotypes via cumulative cultural evolution. Psychological Science, 2014.
>
> [5] Bansak et al., Improving refugee integration through data-driven algorithmic assignment. Science, 359(6373):325–329, 2018.
>
> [6] He, Joyce C., et al. "Stereotypes at work: Occupational stereotypes predict race and gender segregation in the workforce." Journal of Vocational Behavior 115 (2019): 103318.

---

> > ### Author Response · Authors · 2025-11-26
> > **Thank you for your review!**
> >
> > Dear Reviewer vqkN,
> >
> > We are very grateful for the feedback and comments you have provided in your review, which has helped us uncover further insights stemming from our core result.
> >
> > As the rebuttal process is ending in just under one week, we would like to know if we have adequately resolved your concerns and questions. If not, we would be more than willing to engage in further discussion.
> >
> > Sincerely,
> >
> > The Authors

---

### Official Review · Reviewer_MUv3 · 2025-11-02

**Soundness:** 3
**Presentation:** 3
**Contribution:** 2
**Rating:** 4
**Confidence:** 4

**Summary:**

The paper investigates emerging biases that LLMs develop in the multi-turn setting. They find that not only existing bias but also emerging bias can be significant issues for real-world applications such as hiring decision-making. They conduct a game of a sequential hiring paradigm following the existing work. Their result reveals that even though four demographic groups actually have the same success rate across jobs, LLMs develop their own biases for each demographic group, similar to human participants. Even LLMs showed bigger biases than humans. Lastly, they test several interventions, such as prompt steering, to reduce the emerging bias.

**Strengths:**

- Investigate emerging biases, which have been underexplored
- Test many models across six families and various schemes such as CoT
- Explore interventions to reduce the emerging biases

**Weaknesses:**

While the paper is well written and offers insights into emerging biases in LLMs, the paper has limited novelty and contribution in my opinion.

The results themselves are straightforward; when only demographic group information is available, models should naturally use that information to maximize their incentives. Providing more information about candidates would have reduced this effect (as demonstrated in the paper), because additional information allows the model to rely on other signals for decision-making. However, this introduces existing biases in the models.

The paper evaluates bias emergence within a single domain-specific scenario (hiring simulation). The task itself is from the existing paper, and the observed biases are easily mitigated using straightforward prompt-steering techniques. These points collectively limit the paper’s overall novelty and contribution.

**Questions:**

Please see weaknesses

---

> ### Author Response · Authors · 2025-11-20
> **Response to Reviewer MUv3 (1/2)**
>
> We thank the reviewer for their helpful comments and for acknowledging the insights that our work offers in understanding emergent biases in LLMs.
>
> > **“The paper evaluates bias emergence within a single domain-specific scenario (hiring simulation).”**
>
> Our original submission had two scenarios, hiring (main paradigm) and refugee resettlement (original Section 5.2 paragraphs 3 & 4). The latter was specifically used to examine the effect of additional features, based on a task in social science [1]. We recognize this could be clearer in writing. To address the reviewer’s concern, we have added a section highlighting both the refugee setting and a new military conscription setting [2], and provide updated writing and results to reflect these three settings (see Section 5.2).
>
> In both new settings, LLMs also developed biased associations under fully symmetric ground-truth conditions, suggesting that our findings are robust across decision contexts. Prompts and details/results for these settings are in Appendices A.4, A.5, and F.
>
> > **“when only demographic group information is available, models should naturally use that information to maximize their incentives.”**
>
> In the hiring paradigm that we borrow from Bai et al., participants are given points for correct assignments, and so it is indeed natural that models should try to maximize their incentives (i.e., points earned). However, in the refugee and new conscription scenario **these point incentives are completely removed, yet we still see strong stratification effects** (see new Section 5.2, in blue). This indicates that these emergent biases develop even when the model only sees whether assignments are successful or not—an even more concerning finding. We also note that the tasks were carefully designed so that demographic group information is not actually useful in maximizing reward – in the basic tasks, all groups have the same probability of success.
>
> It is also worth noting that in the original hiring paradigm, LLMs stratify much more severely than humans (Figure 2). This is also exacerbated with newer and larger models (Figure 3), in contrast to improved performance on benchmarks that measure existing bias such as BBQ [3]. **One of the main goals of our work is to draw attention to how much the same incentive leads to different levels of stratification across LLMs and humans.** Without the mitigation procedures that we explore in our paper, **real deployment of these systems could lead to more serious harm to any demographic group in the future.**
>
> We thank the reviewer for these two comments, as it has led to new valuable experiments and improved framing and writing.
>
> > **“Providing more information about candidates would have reduced this effect (as demonstrated in the paper), because additional information allows the model to rely on other signals for decision-making. However, this introduces existing biases in the models.”**
>
> We believe that while it is intuitive that increasing the number of features would decrease stratification, the same can be said about all other interventions we test—from CoT to temperature to changing success rates to prompt steering. Part of the contribution of our paper is comparing these different approaches, identifying which work and which do not, as well as the strengths and limitations of each.
>
> Our analysis is thorough for each intervention; for example, we test multiple types of features and find that salient ones such as age/education are much more effective at reducing stratification (average SI reduction = 0.56) than incidental observational cues like hair color (average SI reduction = 0.31). **Because feature salience can vary in different deployment settings, these divergent outcomes underscore that models do not rely on other signals for decision-making in a predictable or straightforward way.**

---

> ### Author Response · Authors · 2025-11-20
> **Response to Reviewer MUv3 (2/2)**
>
> > **“the observed biases are easily mitigated using straightforward prompt-steering techniques.”**
>
> Similar to the previous point, we believe that this misrepresents our contributions. Throughout Section 5, we tested a variety of potential interventions aimed at encouraging the LLM to create more diverse allocations. These include:
>
> - temperature (5.1)
> - chain-of-thought (5.1)
> - changing the setting and removing explicit incentives (5.2)
> - varying success probability rates: both static values and elicited priors (5.2)
> - providing additional information: sets of both salient and spurious features (5.2)
> - four steering prompts that target different aspects of fairness (5.3)
>
> All of these could potentially lead to reduced stratification, and a key contribution of our paper is testing each method to determine which actually do work, and what limitations they have.
>
> In particular, **only one prompt steer out of the four we tried reliably alleviated model tendencies for segregation.** This prompt involved directly incentivizing the model to be diverse for a monetary bonus. This was the only intervention that reduced stratification to the random baseline consistently, strongly suggesting that emergent biases are not easily mitigated nor straightforward to erase. Specifically, **it suggests that only intervening on the objective function specified for models is likely to be successful.**
>
> Furthermore, in Appendix H, we demonstrate the limitations of this prompt intervention by applying it in an artificial setting where demographics are artificially more successful at performing some job categories than others. In this setting, we observed noticeably lower overall success rates with the steering prompt compared to without (31.3% vs. 50.8%). Since the ideal balance between fairness and optimality is often context sensitive, **this dilemma prevents this steering approach from serving as a general mitigation strategy.**
>
> > **“These points collectively limit the paper’s overall novelty and contribution.”**
>
> We hope that through our response, we have addressed the reviewer’s concerns about domain-specificity and contribution. Please feel free to follow up with any further questions.
>
> &nbsp;
>
> [1] Bansak et al. Improving refugee integration through data-driven algorithmic assignment. Science, 2018.
>
> [2] Sørlie et al. Person-organization fit in a military selection context. Military Psychology, 2020.
>
> [3] Parrish, Alicia, et al. "BBQ: A hand-built bias benchmark for question answering." Findings of the Association for Computational Linguistics: ACL 2022.

---

> > ### Comment · Reviewer_MUv3 · 2025-11-22
> >
> > I appreciate the authors’ great efforts, and sorry for missing the additional refugee resettlement scenario in the original submission.
> >
> > However, I still lean toward rejection. My main concern remains that the results are conceptually straightforward.
> >
> > Even though the reviewer said "However, in the refugee and new conscription scenario these point incentives are completely removed, yet we still see strong stratification effects (see new Section 5.2, in blue).", I don't think it's correct. When LLMs are given the tasks, their goals is "maximizing the success" (I also check the prompt that the authors used for the game settings). Given the incentive (here, the incentive means achieving the goal), the models should strategically use the information available to them. Actually, if there were truly no incentive at all, the models would behave essentially like random functions. The phenomenon the paper refers to as "emergent bias" is mathematically trivial: based on a small set of early observations, the models naturally exhibit sample bias.
> >
> > From this standpoint, it is natural that LLMs show more bias than humans; LLMs often behave in more optimized, goal-directed ways, and larger LLMs more so than smaller ones. Although the authors said that their results contrast with previous findings (e.g., existing benchmark results), recent papers have also reported that larger LLMs exhibit more pronounced fairness issues. I suspect that in many existing benchmarks, larger models simply refuse to answer, which artificially makes them appear "fairer."
> >
> > Overall, I'm not convinced that the phenomenon observed in this paper is particularly concerning. If we changed the game setup so that the four demographic groups had different actual success probabilities (sadly it's true in many real-world contexts), what strategy would a model adopt to maximize success? If the model doesn't show any of the"emergent bias", which the paper refer to, the model will fail in the new situation.
> >
> > The setup that I find is more interesting would be that the models show preference to a specific demographic group even when "they know that all groups have equal capability to succeed."

---

> ### Author Response · Authors · 2025-11-26
> **Thank you for your continued engagement (1/2)**
>
> We thank the reviewer for their continued attention to our paper. We first address the third point with additional experiments, before circling back to points 1 and 2.
>
> > **“If we changed the game setup so that the four demographic groups had different actual success probabilities [...] If the model doesn't show any of the ‘emergent bias’, which the paper refers to, the model will fail in the new situation.”**
>
> We run a new experiment testing when demographic groups perform differently at different job classes. For each job category, the four demographic groups have success rates of 0.9, 0.85, 0.8, and 0.75, assigned randomly. For equality, each group was the best at one job type, second best at another, etc. We run the experiment and after the hiring rounds, we ask the LLM “Which group do you think is most likely to succeed at the job of {X}?” for each job.
>
> Averaging across all 40 jobs, 2 models, direct and CoT prompts, and n=30 runs per setting, we find that the LLM only identifies the best-performing group 27.3% of the time, barely surpassing random chance. It mistakenly identifies the second-best group 21.4% of the time, the third best group 21.5% of the time, and even the worst-fitting group 29.8% of the time. This means that 72.7% of the time, models are incorrectly biased against the best-performing group. Experiments were run on Gemini 2.5 Flash and GPT-4o.
>
> | Model | Prompting | % of times best selected | 2nd-best  | 3rd | worst |
> |-|-|-|-|-|-|
> | **GPT-4o** | CoT | 0.28 | 0.35 | 0.18 | 0.20 |
> | | direct | 0.28 | 0.20 | 0.26 | 0.27 |
> | **Gemini 2.5 Flash** | CoT | 0.27 | 0.30 | 0.19 | 0.24 |
> | | direct | 0.23 | 0.31 | 0.23 | 0.23 |
>
> **Here, our findings are no longer trivial: LLM are actively under-exploring, directly leading to impressions of groups that are both incorrect and harmful.** Moreover, this illustrates how LLMs are ill-equipped to handle these multi-turn exploration tasks where early sample bias can create stereotypes about groups.
>
> We also performed the same test for 80 rounds, explicitly telling the LLM it had a longer time horizon. However, there was no statistically significant difference vs. the 40-round case (26.2%, 30.5%, 24.4%, 18.9%), suggesting the inability of LLMs to appropriately adapt their exploration in settings that allow for more exploration to attain a better long-term reward.
>
> | Model| Prompting | % of times best selected | 2nd-best  | 3rd | worst|
> |-|-|-|-|-|-|
> | **GPT-4o** | CoT | 0.28 | 0.30 | 0.33 | 0.10 |
> | | direct | 0.24 | 0.23 | 0.32 | 0.21 |
> | **Gemini 2.5 Flash** | CoT | 0.26 | 0.39 | 0.14 | 0.21 |
> | | direct | 0.27 | 0.30 | 0.18 | 0.25 |
>
> We have included a writeup of this new experiment and related results in Appendix K in the newest revision. While sample bias itself may be “mathematically trivial”, the strategy to select samples certainly is not, and this is what we demonstrate that LLMs lack in practice: they under-explore and form judgments too quickly, leading to suboptimal (and harmful) social outcomes.

---

> ### Author Response · Authors · 2025-11-26
> **Thank you for your continued engagement (2/2)**
>
> > **“point incentives are completely removed”**
>
> We clarify that our target was to remove any additional individual incentives (points, bonus pay) that the LLM would get outside of following the user’s instructions. We agree that with the reviewer’s definition “incentive means achieving the goal”, there is always going to be incentives. We have further revised our writing in Section 5.2 with regard to this (orange text).
>
> > **“it is natural that LLMs show more bias than humans; LLMs often behave in more optimized, goal-directed ways”**
>
> Yes, a central theme our paper seeks to highlight is that while LLMs have gotten better at optimizing towards particular objectives, this doesn’t necessarily lead to better outcomes overall. Instead, **it becomes more important to define more detailed and desirable objectives for socially sensitive tasks**. This is illustrated through our series of interventions, where only introducing a socially nuanced objective function robustly fixes the emergent stratification behaviors we see. We discuss this point in the last paragraph of our introduction.
>
>
> > **“I suspect that in many existing benchmarks, larger models simply refuse to answer, which artificially makes them appear "fairer.""**
>
> The BBQ benchmark [1] that we use in Appendix D has its evaluation protocols (as implemented by HELM) structured in a way that does not permit for refusals to count as a “fair response” due to the multiple-choice nature of the responses. As a result, improvements on BBQ with scale cannot be attributed to refusal behaviors. We believe this contrast helps contextualize why emergent biases in multi-turn settings represent a distinct and understudied failure mode.
>
> Furthermore, while “recent papers have reported that larger LLMs show more fairness issues,” none of them predict that models will invent new biases with no basis in the training distribution. This is particularly true in fairness research, where such a mechanism of emergent bias formation has not been proposed. This matters because it points to a gap in current interventions: solutions such as post-hoc prompts (“don’t be biased”), rebalancing datasets, projecting out bias dimensions, etc. [2] either do not apply to emergent biases or do not reliably remove them. **Our work represents the first demonstration, to our knowledge, of both invented biases and the use of incentives to mitigate them, a solution empirically selected among numerous candidate interventions.**
>
> Lastly, we appreciate the reviewer sharing the setup they find interesting. However, our impression is that if “[models] know that all groups have equal capability to succeed,” then the models would have no opportunity to develop biases at all. If our current new experiments do not suffice, we ask the reviewer to please clarify further.
>
> [1] GPT-5 System Card, OpenAI.
>
> [2] Gallegos et al. Bias and Fairness in Large Language Models: A Survey. Computational Linguistics, 2024.

---

### Author Response · Authors · 2025-11-20
**General Response**

We thank the reviewers for each providing constructive and insightful feedback for our paper!

In particular, we appreciate that all three reviewers noted that our work investigates underexplored/novel questions, and that reviewer vqkN specifically acknowledged our paper as very well-motivated and engaging with a broad literature.

In our rebuttals, **we provide a tailored response to each weakness and question from each reviewer, backed up by new experiments and changes to the pdf** (marked in blue).

We also provide a list of key changes and invite reviewers to peruse them at their convenience:
- Domain generality: new experiments in military conscription and refugee reallocation settings. (MUv3 point 1)
&nbsp;
- Removal of gamified incentives: these new settings do not contain points for successful assignments. (MUv3 point 2)
&nbsp;
- Generalization to agentic systems: ReAct framework (vqkN point 1)
&nbsp;
- Comparison between stratification and BBQ performance: strong opposite trend (vqkN point 2)
&nbsp;
- Real demographics: new experiment testing socially salient (White, Asian, Black, Hispanic) demographics (vqKN point 5)
&nbsp;
- Expanded human participant information (kwFn point 1)
&nbsp;
- Metric justification: equivalence proof of Mutual Information vs. Stratification Index (kwFn point 2)
&nbsp;
- Metric explanation: BGD, GASI, and JSD (kwFn point 2)
&nbsp;
- Metric convergence under large sample sizes (kwFn point 3)
&nbsp;
- Interpretation for Appendix E (old Appendix B): Visualization of model differences (kwFn point 4)
&nbsp;
- Expanded temperature generality: T=0 and T=1.5 across frontier models (kwFn point 6)
&nbsp;

To all reviewers, please feel free to respond with any questions or comments.

Sincerely,

Authors

---

### Meta-Review · Area_Chair_JH9F · 2026-01-04

**Summary:**

1. Reviewer MUv3

The main concerns are:

 (1) The motivation and the results are straightforward.

 (2) The paper evaluates bias emergence within a single domain-specific scenario (hiring simulation).

2. Reviewer vqkN

The main concerns are:

(1)  if improved in-context learning is why better models stratify more

(2) a new plot comparing results with existing benchmarks

(3) if the artificial setting causes the LLM to act without constraints that would apply to real demographics

(4) if “spurious features” in an ablation are indeed spurious

(5) how our results interact with pre-existing bias for real demographics.

3. Reviewer kwFn

(1) The authors compare the simulation results of LLMs with those from human participants, but lack descriptions of the human participants, such as the sample size and distribution of demographic variables.

(2) need to provide more explanation for the three newly defined metrics.

(3) repeated individuals’ information in many rounds

(4) how changing model temperature affects the degree of stratification

(5) if the human data band in Figure 2 is standard deviation

(6) if emergent biases were revealed in multi-turn interactions even before LLMs

**Reviewer Concerns:**

1. Reviewer MUv3

The first point on the novelty has not been addressed.

2. Reviewer vqkN: Most of the concerns have been addressed.

3. Reviewer kwFn:

(1) There is no human results for the new added settings.

(3) This has not been addressed

(6) this problem has not been addressed, the authors should add experiments on classical machine learning models or deep learning models.

**Reviewer Scores:**

Reviewer MUv3 will not change the score

Reviewer kwFn will not change the score

---

### Decision · Program_Chairs · 2026-01-26

Reject